# Proteomic Analysis of Zeb1 Interactome in Breast Carcinoma Cells

**DOI:** 10.3390/molecules26113143

**Published:** 2021-05-24

**Authors:** Sergey E. Parfenyev, Sergey V. Shabelnikov, Danila Y. Pozdnyakov, Olga O. Gnedina, Leonid S. Adonin, Nickolai A. Barlev, Alexey G. Mittenberg

**Affiliations:** 1Institute of Cytology of the Russian Academy of Sciences, 194064 St. Petersburg, Russia; gen21eration@gmail.com (S.E.P.); buddasvami@gmail.com (S.V.S.); 9apdu179@gmail.com (D.Y.P.); olga.o.gnedina@gmail.com (O.O.G.); nick.a.barlev@gmail.com (N.A.B.); 2Moscow Institute of Physics and Technology, 141700 Dolgoprudny, Russia; leo.adonin@gmail.com

**Keywords:** breast cancer, Zeb1, metastasis, epithelial to mesenchymal transition (EMT)

## Abstract

Breast cancer is the most frequently diagnosed malignant neoplasm and the second leading cause of cancer death among women. Epithelial-to-mesenchymal Transition (EMT) plays a critical role in the organism development, providing cell migration and tissue formation. However, its erroneous activation in malignancies can serve as the basis for the dissemination of cancer cells and metastasis. The Zeb1 transcription factor, which regulates the EMT activation, has been shown to play an essential role in malignant transformation. This factor is involved in many signaling pathways that influence a wide range of cellular functions via interacting with many proteins that affect its transcriptional functions. Importantly, the interactome of Zeb1 depends on the cellular context. Here, using the inducible expression of Zeb1 in epithelial breast cancer cells, we identified a substantial list of novel potential Zeb1 interaction partners, including proteins involved in the formation of malignant neoplasms, such as ATP-dependent RNA helicase DDX17and a component of the NURD repressor complex, CTBP2. We confirmed the presence of the selected interactors by immunoblotting with specific antibodies. Further, we demonstrated that co-expression of Zeb1 and CTBP2 in breast cancer patients correlated with the poor survival prognosis, thus signifying the functionality of the Zeb1–CTBP2 interaction.

## 1. Introduction

Breast cancer (BC) is the most commonly diagnosed malignant tumor in humans. More than 50,000 new cases of breast cancer are recorded annually in Russia, and in the whole world, the number exceeds 1 million [1,2,3]. Breast carcinomas are also considered the second leading cause of cancer death in women [4]. Among patients, more than 90% of deaths associated with breast cancer are caused not by the primary tumor, but by metastases. In 6–10% of breast cancer diagnoses, the tumor has already metastasized to other sites, and in 30% of patients with early stages of breast cancer, metastatic or recurrent disease is observed. Understanding the mechanisms of metastasis of this tumor is important for the early diagnosis and treatment of breast cancer. The exact mechanisms involved in the transition of non-invasive tumor cells to those with metastatic potential are still not fully understood. However, recent literature data indicate that one of the main mechanisms of breast cancer metastasis is the epithelial–mesenchymal transition [5].

The epithelial–mesenchymal transition (EMT) is a reversible genetic program implemented during embryonic development and pathologically activated in cancer [6,7]. This process is accompanied by a loss of intercellular contacts and polarity of epithelial cells, a global reorganization of the cytoskeleton, leading to a loss of epithelial properties and an increase in mesenchymal morphology, as well as increased cell motility and resistance to therapy [8,9]. The EMT phenomenon has been examined in the context of distant metastases in different types of carcinomas, including breast tumors. The EMT program is controlled by several groups of transcription factors (EMT-TF), among which the SNAIL1, Zeb1, and Zeb2 master regulators are distinguished [10,11,12]. Their expression correlates with a poor prognosis in patients with breast cancer. The mechanisms of regulation of EMT transcription factors have been actively studied in recent years.

A hallmark of EMT is the attenuation of the expression of E-cadherin, a key cell adhesion molecule. The vast majority of known signaling pathways are involved in the regulation of EMT. Several transcription factors, including the snail/slug, twist, Zeb1/Zeb2, and E12/E47 families, respond to these signals and act as major molecular regulators of the EMT program. These transcription factors recognize E-box DNA sequences located near the transcription initiation sites of the E-cadherin genes, where they recruit cofactors and histone deacetylases [2,3].

The ZEB family (Zeb1/2) is represented by zinc finger transcription factors that recognize the E-box type consensus element, which are known as Zeb proteins. These transcription factors recognize the 5’-CACCT-3 ‘sequences in the promoter regions of various genes and, by binding to them, suppress the transcription of these genes [5]. Expression of Zeb1 and/or Zeb2 increases the aggressiveness and metastatic capacity of breast malignancies. Zeb1 and Zeb2 suppress the expression of not only E-cadherin, but also other epithelial markers involved in cell polarity, tight junctions, gap junctions, and desmosomes. In addition, Zeb1 plays an important role in tumor progression and poor clinical outcome in cancer patients. Zeb1 is a specific inducer of EMT, which ensures the manifestation of such properties as radioresistance and drug resistance by cancer stem cells. Zeb2 directly inhibits the expression of the tight junction proteins claudin-4 and zona occludens 3. It also reduces the expression of the desmosomal protein plakophilin-2 and induces the expression of mesenchymal proteins, such as vimentin, N-cadherin, and matrix metalloproteinase-2 through an unknown mechanism.

During the EMT process, cells lose epithelial traits and acquire mesenchymal characteristics. EMT is characterized by loss of cell adhesion and phenotypic change from a typical cubic to an elongated spindle-like shape, which leads to an increase in the migration capacity of the cells. At an early stage of tumor metastasis, cancer cells from the primary tumor node can acquire invasive properties and gain access to blood or lymphatic vessels as circulating tumor cells, which is ensured by neoangiogenesis and basement membrane remodeling. In the blood and lymphatic vessels, circulating tumor cells are able to survive and eventually reach distant secondary sites such as bones, lungs, liver, and brain [13,14]. This occurs mainly during the mesenchymal–epithelial transition (MET), which makes a significant contribution to the colonization of circulating tumor cells of metastatic tumors at the secondary site. These dynamic EMT/MET state transitions can play a decisive role in tumor metastasis. The EMT/MET processes should not be compared to a switch, but rather to extremely flexible process of transdifferentiation from one state to another and vice versa, recently referred to as “epithelial–mesenchymal plasticity”. In recent years, several research groups have identified hybrid epithelial–mesenchymal phenotypes with a “partially” activated EMT program, which supports this concept. In the course of tumor progression, it contains multiple subpopulations of cells characterized by different EMT states and different invasive and metastatic potential [15,16].

Such complex metastable states of the cell require no less subtle and sophisticated multilevel tuning, including the molecules that regulate these states. This paper investigates the interactome of Zeb1, a key transcription factor that plays a central role in the management of the epithelial–mesenchymal transition.

## 2. Results and Discussion

### 2.1. EMT Induction

According to the literature (see review [17]), the increased Zeb1 expression was shown suffice to trigger the epithelial–mesenchymal transition (EMT). In this regard, we decided to ectopically express Zeb1 in Zeb1-negative luminal breast cancer epithelial cell line, MCF-7. To this end, the expression of the Zeb1/GFP fusion protein, which was under control of Tet-inducible operator, was activated by adding doxycycline to the culture medium The expression efficiency was detected by GFP fluorescence (Figure 1a). Importantly, activation of Zeb1 expression in MCF-7 cells was sufficient to induce phenotypic changes characteristic of EMT (Figure 1b). To estimate the EMT phenotype acquisition we performed a wound-healing assay, which demonstrated the increase of migration ability of Zeb1-expressing cells (Figure 1c,d).

In the experiments, MCF-7/Zeb1 cells without Zeb1 induction and at 24 and 72 h after the addition of doxycycline were used, which corresponds to the absence, early and late stages of EMT, respectively. In control MCF-7/Zeb1 cells without Zeb1 induction no GFP fluorescence was observed (Figure 1a). After the doxycycline addition to the MCF-7/Zeb1 cell culture, a gradual increase in the intensity of GFP fluorescence in the cell nuclei was observed, which indicates that the Zeb1/GFP fusion protein was induced (Figure 1a). However, GFP fluorescence can only indirectly confirm the induction of Zeb1 expression and hence initiation of EMT; therefore, to additionally control the activation of the EMT-like transition, we also used real-time PCR and Western blot analysis of molecular markers of EMT.

The epithelial–mesenchymal transition is finely regulated by a variety of signaling pathways. Induction of Zeb1 leads to a change in the expression pattern of many epithelial and mesenchymal markers, which are responsible for changes in cell morphology and their properties. To confirm the initiation of EMT, we performed a Western blot analysis of epithelial marker E-cadherin (Figure 2a). To correctly assess of changes in the content of macromolecules of our interest, the amounts of protein in the samples (with or without EMT induction) were normalized to the content of actin (Figure 2a). It is well established that the induction of EMT downregulates the epithelial marker E-cadherin, one of the most important cell adhesion molecules. The Zeb1 EMT-TF, acting on the promoter elements of the E-cadherin gene, causes a decrease of its expression and serves as the most important stimulus for EMT [18]. Immunoblotting revealed a gradual decrease in the amount of E-cadherin, as well as differences in the electrophoretic mobility of the protein from both control cells and cells with induced EMT, which may be associated with the appearance of another isoform of E-cadherin (Figure 2a). According to the literature, the Zeb1 TF is able to induce alternative splicing of the CDH1 gene, leading to the appearance of a non-functional E-cadherin protein [19]. Based on the data obtained, one can conclude that changes in E-cadherin content indicate the induction of EMT-like state in MCF-7 cells, which originally are epithelial. However, it should be noted that when we attenuated the expression of Zeb1 by specific Zeb1-shRNA in mesenchymal-like triple-negative breast cancer cell line, MDA-MB-231, these cells did not acquire epithelial phenotype (data not shown). These results suggest that apparently, besides Zeb1, other transcription factors (e.g., p53) may restrict the phenotypic plasticity of cancer cells.

Western blot results demonstrated the presence of Zeb1 protein only in the cells after doxycycline induction, thus additionally confirming the activation of the Zeb1/GFP fusion protein expression. The presence of several bands on the immunoblot corresponding to the Zeb1 protein is explained by the presence of several post-translationally modified isoforms of the Zeb1/GFP fusion protein in the cells (Figure 2a).

Also, EMT induction in MCF-7/Zeb1 cells was confirmed by real-time PCR with oligonucleotide primers specific to Zeb1, E-cadherin, N-cadherin, and vimentin. The observed decrease in E-cadherin (Figure 2b) and gradual increase in Zeb1 (data not shown), N-cadherin and vimentin (Figure 2b) expression served as an additional corroboration of the EMT-like induction in the present cell model.

ZEB1 is known to repress ER-α expression by forming a DNMT3B-containing complex on its promoter. The latter induces DNA hypermethylation and hence, attenuation of ER-α expression. Notably, the downregulation of ZEB1 was shown to restore ER-α activity thereby increasing the sensitivity of breast cancer cells to antiestrogen treatment [20]. In agreement with these data, we have also shown that ectopic expression of Zeb1 significantly attenuated the ER-α gene expression; hence, confirming the EMT-associated changes in MCF-7 cells upon the induction of Zeb1.

### 2.2. Isolation and Analysis of Zeb1 Interactome

To isolate the Zeb1 interactome in nuclear extracts of MCF-7/Zeb1 cells after EMT induction, co-immunoprecipitation with llama nanoantibodies to the GFP protein immobilized on sepharose was used. Since control MCF-7/Zeb1 cells without doxycycline induction lack GFP, any protein binding by nanoantibodies would be considered as non-specific.

Subsequent mass spectrometric analysis has identified 177 confident proteins (Appendix A). In the control MCF-7 samples without the Zeb1/GFP insert, the results of mass spectrometric analysis revealed about one hundred proteins to be nonspecifically bound to GFP antibody (data not shown). Since these proteins were found in the rest of the samples, they were excluded from further analysis. In the control MCF-7/Zeb1 cells before the induction, 141 proteins were found, most of which were nonspecifically bound; for this reason, they were also excluded from the analysis, however, due to the possible low activation of Zeb1 expression because of the leakiness of the system, some of the identified proteins were taken for consideration in subsequent analysis. In the cell samples after at 24 h and 72 h of Zeb1 induction, 120 and 69 proteins were identified, respectively. These differences in the numbers of interacting proteins may indicate that the interactome of Zeb1 factor at the early and late stages of EMT is different. For further analysis, after excluding the nonspecifically bound ones, only those proteins identified by mass spectrometry in at least two replicates were used. In addition, according to the results of mass spectrometry, we have reliably demonstrated the presence of Zeb1 and GFP proteins only in the cell samples with induced EMT, which confirmed the assumption about the non-specificity of protein binding in control samples.

The STRING analysis [21] identified a number of proteins that have been demonstrated previously to interact with Zeb1 (Figure 3). Further, we analyzed the potential role of the selected proteins in metastasis and development of breast cancer, as well as their effect on the functioning of the Zeb1 EMT-TF. As a result, several proteins have been identified that can be potential targets for affecting the functions of Zeb1.

#### 2.2.1. RNA Helicase DDX17

We consider the ATP-dependent RNA helicase DDX17 as one of such proteins. According to the literature, the expression of this protein, is increased in several types of tumors, including breast cancer [21]. The data available on the structure-functions features of DDX17 made this protein an interesting candidate for further analysis. This protein was reliably identified in control cells without EMT induction and in cells with induced EMT; however, we noted an increase in the content of this enzyme in experimental samples, especially at the late stage of Zeb1 activation (72 h), which requires further confirmation. It has been established that this protein can also act as a cofactor for several transcription-related factors, such as the alpha-estrogen receptor, beta-catenin, and p53 tumor suppressor, and thus play a significant role in the formation of malignant neoplasms, including breast cancer [22,23]. In addition, the interaction of this enzyme with the zinc finger protein ZAP suggests the possibility of direct interaction of DDX17 with Zeb1 [24]. This protein also plays the role of a regulator for the YAP factor [25], which, in turn, is able to activate the expression of genes whose products are involved in the epithelial–mesenchymal transition, including Zeb1. In addition, the YAP protein is also a known coactivator of the Zeb1 factor that participates in malignant cell transformation [26].

#### 2.2.2. C-Terminal-Binding Protein 2

In accord with previously published results [27,28,29], we identified the CTBP2 protein as an interacting partner of Zeb1. Its ability to directly interact with Zeb1 in the neuronal progenitor cells has been shown in the literature [28]. In addition, this protein is involved in the regulation of the proliferation and migration of breast cancer cells by acting on the inhibitor of cyclin-dependent kinases p16INK4A. The latter, in turn, is regulated by the transcription factor Zeb1 [30], and is also directly involved in EMT of hepatocellular carcinoma cells [31]. In addition, CTBP2 plays a role in promoting drug resistance in cancer cells [32]. Recent data found CTBP2 to cause metastasis via activation of TGF-β signaling [33].

### 2.3. Mass Spectrometry Data Verification

The presence of the selected interactants (DDX17, MPG, CTBP2) was confirmed by immunoblotting (Figure 4), which allowed us to assess the changes in the content of the studied proteins at different stages of EMT and we made sure that the mass spectrometry data were consistent with the immunoblotting results.

The detection of the analyzed CTBP2 protein both at the early and at the late stages of Zeb1 activation suggests the participation of this factor in the entire process of epithelial–mesenchymal transition. In turn, an increase in the DDX17 RNA helicase content was observed only at 72 h after Zeb1 induction. The epithelial–mesenchymal transition is a multi-stage process, each part of which requires the action of various signaling pathways. Such proteins can participate in the EMT completion. The previously noted decrease of vimentin expression at the late stages of EMT may be associated with a change in the morphofunctional status of the cell at the final stages of the epithelial–mesenchymal transition.

Recent reports suggest that the C-terminal binding protein 2 (CtBP2) is a critical transcriptional co-repressor that orchestrates the EMT program both in Zeb1-dependent and -independent ways [34,35]. Importantly, CTBP2 may serve as an independent prognostic marker for several tumors including hepatocellular carcinoma and lung cancer [36,37]. To address the importance of Zeb1–CTBP2 interaction for carcinogenesis we have performed the bioinformatic analysis of possible correlation between the expression levels of CTBP2 and survival outcomes among breast cancer patients with either high or low expression of Zeb1 (Figure 5). Our results suggest that the life expectancy of patients with high expression of both Zeb1 and CTBP2 was shorter compared to that of patients with low expression of both genes. This difference in survival was observed only in Zeb1-positive patients (Figure 5). Thus, our new data support the functional significance of the Zeb1–CTBP2 interaction.

## 3. Materials and Methods

### 3.1. Cell Culture

The culture of human breast adenocarcinoma cells MCF-7 [38] with lentiviral insertion of the Zeb1 gene fused with the GFP gene, under a tetracycline-activated promoter [39], was used. The cells were cultured according to a standard protocol in a DMEM medium with 10% fetal bovine serum, 0.06% L-glutamine, 0.001% insulin and 0.004% gentamycin in an incubator in atmosphere of 5% CO2 at 37 °C.

To induce the expression of the Zeb1/GFP fusion protein, doxycycline at a final concentration of 0.5 μg/mL was added to the culture medium. Expression was detected by GFP luminescence using a Zoe fluorescence microscope (Bio-Rad, Basel, Switzerland).

### 3.2. Wound Healing Assay

MCF7/Zeb1 cells were seeded into the 48-well plates with DMEM medium supplemented with 10% FBS and pen/strep. Zeb1 had being activated with doxycycline for 24 and 72 h. Linear scratches were made with 200 μL tips on a cell monolayer at 80% confluence. The overgrowth of the wound was photographed 0 and 24 h after generation of the scratch using an FV3000 confocal microscope (Olympus, Shinjuku, Tokyo, Japan).

### 3.3. Co-Immunoprecipitation

To isolate the ZEB1 interactome the MCF-7 nuclear extracts were co-immunoprecipitated with llama nano-antibodies to green fluorescent protein immobilized on sepharose (Abcam ab191863, Cambridge, UK). The protein conjugates immunoprecipitated on sepharose beads were washed four times with RIPA buffer followed by dissolving in Laemmli buffer and subjected onto SDS–PAGE to remove PEG derivatives present in wash buffer.

### 3.4. Sample Preparation and LC-MALDI Mass Spectrometry

The protein samples were in-gel digested overnight by Trypsin Gold (Promega, Madison, WI, USA) followed by extraction and drying in Martin Christ RVC-2-33IR rotary vacuum concentrator (Martin Christ, Germany). Prior to reversed-phase fractionation, digested samples were resuspended in 50 μL of 1% (*v*/*v*) formic acid in water and filtered through 0.2 μm PVDF filter. Peptides were separated with a Chromolith CapRod RP-18e HR reversed-phase column (0.1 mm × 150 mm, Merck, Darmstadt, Germany) on a nano LC system (Eksigent NanoLC Ultra 2D+ system, SCIEX, Darmstadt, Germany). A total peptide amount of 600 ng was loaded and separated using a linear gradient of 0–50% B over 115 min followed by 50–100% B for 1 min and 100–100% B for 4 min at a flow rate of 400 nl/min. The mobile phases used were A, 5% acetonitrile with 0.2% (*v*/*v*) TFA in water and B, 60% (*v*/*v*) acetonitrile in water. The column was operated at a room temperature of 22–24 °C. The effluent from the column was mixed with matrix solution (CHCA 5 mg/mL, 0.2% (*v*/*v*) TFA in 95% methanol) containing two calibration standards bradykinin 2–9 (30 pM/mL) and ACTH 18–39 (60 pM/mL), at a flow rate of 1.4 μL/min. A micro-fraction collector was used to deposit 1 mm spots every 5 s, and a total of 1408 fractions were collected in a 44 × 32 array for each nano LC run. The column was washed with a gradient (0–100–100% B for 5 min and 2 min, respectively, at a flow rate of 800 nl/min) and equilibrated to 0% B for 3.5 min before subsequent injections.

The fractionated samples were analyzed with a TOF/TOF 5800 System (SCIEX, Darmstadt, Germany) instrument operated in the positive ion mode. The MALDI stage was set to continuous motion mode. MS data was acquired at 2600 laser intensity with 1000 laser shots/spectrum (200 laser shots/sub-spectrum) and MS/MS data were acquired at 3400 laser intensity with a DynamicExit algorithm and a high spectral quality threshold or a maximum of 1000 laser shots/spectrum (250 laser shots/sub-spectrum). Up to 25 top precursors with S/N > 40 in the mass range 750–4000 Da were selected from each spot for MS/MS analysis.

The Protein Pilot 5.0 software (SCIEX, Darmstadt, Germany) with the Paragon algorithm 5.0 in thorough mode was used for the MS/MS spectra search against the UniProt human database. Carbamidomethyl cysteine was set as a fixed modification. False discovery rate (FDR) analysis was done by analysis of reversed sequences using the embedded PSEP tool. The mass spectrometry proteomics data have been deposited at the ProteomeXchange Consortium via the PRIDE partner repository with the data set identifier (submitted).

### 3.5. Western Blotting

After separation in SDS–PAGE (AA: 10%, AA/BisAA ratio: 36:1), the proteins were transferred onto PVDF membranes following overnight incubation with specific primary antibodies. The following primary antibodies were used: anti-actin (Sigma A5441, St. Louis, MO, USA), anti-E-cadherin (BD Biosciences, Heidelberg, Germany), anti-Zeb1 (Sigma AMAb90510), anti-CTBP2 (Abcam ab128871), anti-DDX17 (Sigma AV41029), anti-vimentin (Santa Cruz sc6260) and anti-N-cadherin (Cell Signaling 14215S). The secondary antibodies were from Sigma: anti-mouse (A9917) and anti-rabbit (A0545). Bound antibodies were visualized using the SuperSignal West Femto Maximum Sensitivity Substrate ECL kit (Thermo scientific, Boston, MA, USA), chemiluminescence was detected using ChemiDoc Touch Imaging System (Bio-Rad).

### 3.6. Real-Time Polymerase Chain Reaction

Total RNA was isolated from cells using a TRI Reagent (Sigma-Aldrich, St. Louis, MO, USA). cDNA was then synthesized by reverse transcription using MMLV RT kit (Eurogen, Moscow, Russia). qPCR mix-HS SYBR master mix (Eurogen, Moscow, Russia) was used for real-time PCR, with GAPDH as the internal control. The primers are listed in Table 1. The total qPCR reaction volume was 20 μL and consisted of 2 μL of cDNA, 0.1 μL of each primer, 4 μL of qPCR mix, and 13.8 μL of ddH2O. The PCR reaction program was as follows: 95 °C for 1 min; 40 cycles of 95 °C for 15 s, 60 °C for 15 s, then heated to 95 °C for 60 s; and cooled to 4 °C for 5 min. To estimate the relative gene expression the ΔΔCt method was used.

### 3.7. Statistical Analysis

Data are presented as mean with standard deviation (SD) or standard error of the mean (SEM) and as median with quartiles when the experiment is repeated at least three times. Statistical significance was analyzed using Student’s *t*-test or Mann–Whitney test.

### 3.8. Bioinformatic Analysis

The gene expression data and the survival information are derived from The Cancer Genome Atlas (TCGA). Kaplan–Meier survival curves were constructed to evaluate the differences in the overall survival time for clinical datasets of breast cancer (clinical cases) with high and low expression levels of ZEB1 and CTBP2. Cases were divided into two cohorts, high vs. low expression, based on the best cut-off values of gene expression (smallest *p*-value). A total of 389 patients were included in those datasets, from which 55 were patients with the high expression level of ZEB1 and CTBP2, 156 cases with low expression level these genes, 79 patients with low ZEB1 expression and high it of CTBP2, and 99 cases with the high expression level of ZEB1 and low CTBP2 expression. For survival analysis of these data and visualization results the survminer R package was used.

## 4. Conclusions

In summary, we have identified several potential interactors of Zeb1 EMT-TF, which can serve as targets for attenuation of breast cancer metastasis. It is necessary to thoroughly study the possible direct and indirect interaction pathways of these proteins with Zeb1, their role in the processes of epithelial–mesenchymal transition and metastasis, as well as potential ways of influencing their activity.

A comparison of Zeb1 interactome described in our study with the ones described in other studies strongly suggest that the composition of Zeb1-interacting proteins depends on the source of cells used in the study. In this respect, our results are novel and important since they describe new Zeb1-interacting proteins in the p53-positive breast MCF7 cancer cells. It should be noted that most of the proteomic studies have been performed on HEK293 cells [40,41]. Importantly, HEK293 cells do not undergo EMT even upon forced expression of Zeb1. On the contrary, in our epithelial breast cancer cell system (MCF-7) ectopic expression of Zeb1 conferred a partially mesenchymal phenotype to the cells, emphasizing the significance of Zeb1 interactions observed. In addition, the interactome of Zeb1 described in the work of Manshouri et al. [40] does not reflect stable physical interactions of Zeb1 but rather describes proteins that are located in the proximity of Zeb1. This is due to the nature of BirA assay that allows the identifications of all biotin-labeled proteins surrounding the BirA-containing chimeric protein of interest. In contrast, our results provide the information on direct stable interactions of Zeb1 in cells.

It is well established that Zeb1 mediates resistance to anti-cancer therapy. To do that, Zeb1 utilizes several distinct mechanisms, depending on the specific tumor and treatment type. These mechanisms can be mediated by various transcriptional targets of Zeb1, which, in turn, are determined by its interactome.

The present work is the first step in elucidation of the Zeb1 role in carcinogenesis. Considering the importance of mutual regulation between Zeb1 and the tumor suppressor protein p53, we have uncovered—for the first time—that the interactome of Zeb1 EMT-TF is strongly affected by p53, which could help in the search for potential targets for breast cancer therapy.

With the accumulation of new data, it becomes apparent that the spectrum of EMT-TFs action during tumor development is much wider than assumed previously. These are pleiotropic transcription factors able to control not only cell plasticity, oncogenic transformation, motility, and metastasis, but also metabolism, as well as resistance to genotoxic stress. Their versatility and multiplicity of functions makes the task of understanding the molecular details of EMT-TF networking rather challenging. Future studies of the mechanisms of action of Zeb1 and other EMT master regulators should yield a better understanding of the ontogenesis of tumors and how the tumors evolve their resistance to therapy. In this respect, proteomic studies, including the present one, of the EMT-TF interactomes will contribute to the development of more precise and less toxic anticancer therapies.

## Figures and Tables

**Figure 1 molecules-26-03143-f001:**
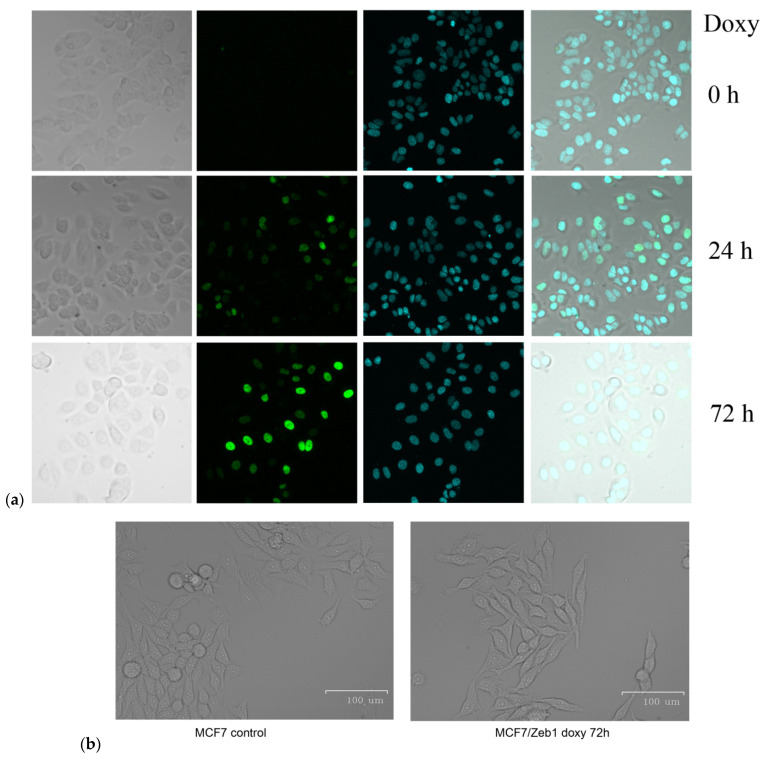
(**a**) Fluorescence of the GFP-Zeb1 hybrid protein in MCF-7 cells at 0 h, 24 h and 72 h after doxycycline induction; (**b**) partial mesenchymal phenotype of MCF-7 cells before (left panel) and after Zeb1 induction (right panel); (**c**,**d**) the wound healing assay (a number of cells populating the scratch) in MCF cells at 0, 24, and 72 h of Zeb1 induction by doxycycline.

**Figure 2 molecules-26-03143-f002:**
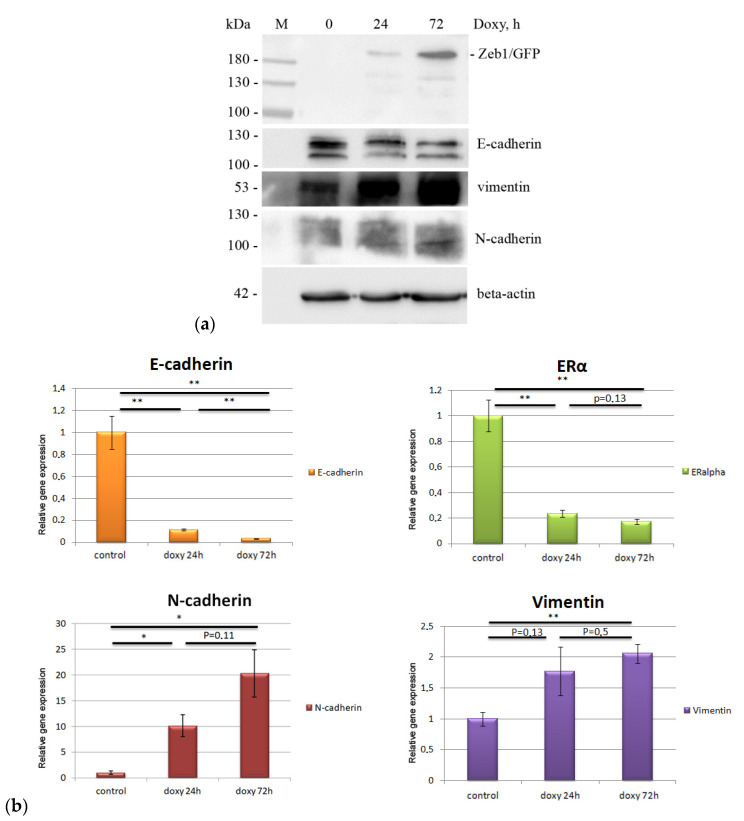
(**a**) Immunoblots of MCF-7/Zeb1 cells total lysates with antibodies to E-cadherin, Zeb1 and beta-actin (samples of MCF-7/Zeb1 cells at 0, 24, and 72 h after doxycycline treatment, respectively). The position of the protein molecular weight marker bands (in kDa) is indicated on the left. (**b**) Real-time PCR analysis of E-cadherin, N-cadherin, ERα and vimentin expression in MCF-7/Zeb1 cells at 0, 24, and 72 h after doxycycline treatment, respectively. The expression levels in control cells were taken arbitrary as 1. *p* > 0.05, *: *p* > 0.01.

**Figure 3 molecules-26-03143-f003:**
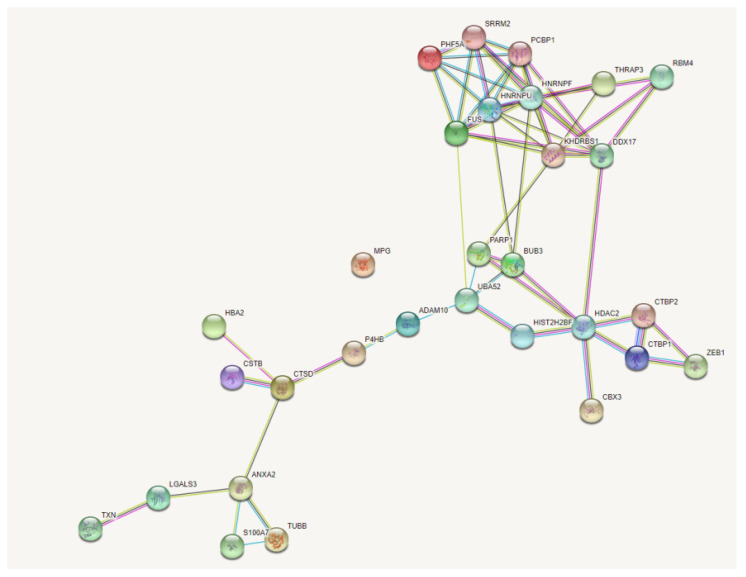
STRING analysis of protein networks, only showing proteins connected to the Zeb1 node.

**Figure 4 molecules-26-03143-f004:**
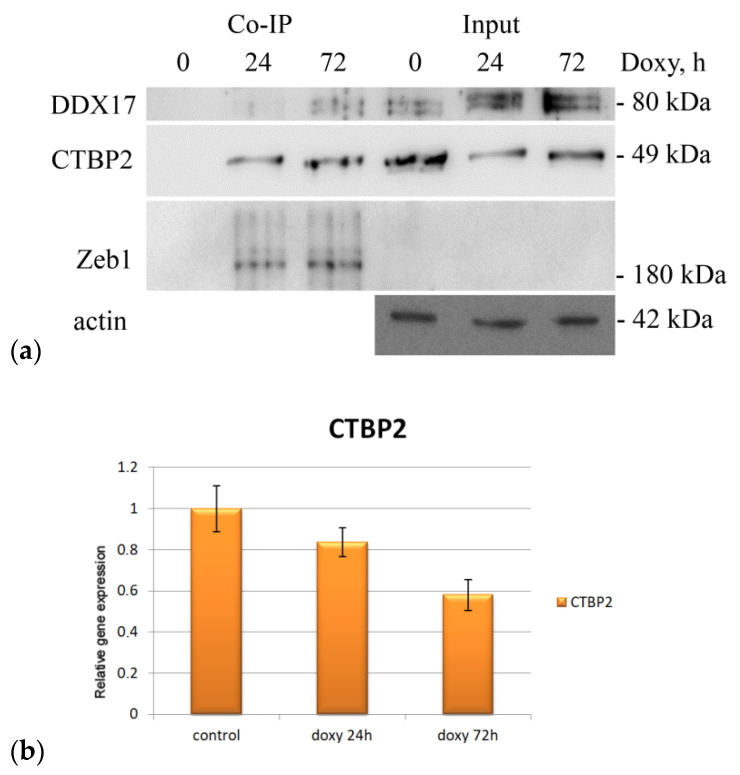
(**a**) Immunoblots of MCF-7/Zeb1 cells nuclear extracts after co-immunoprecipitation with llama nanoantibodies to GFP and inputs (samples of MCF-7/Zeb1 cells at 0, 24, and 72 h after doxycycline treatment, respectively). Primary antibodies to: DDX17, CTBP2 (upper panel); immunoblots of MCF-7/Zeb1 cells nuclear extracts with antibodies to Zeb1 and beta-actin (samples of MCF-7/Zeb1 cells at 0, 24, and 72 h after doxycycline treatment, respectively) (lower panel). The position of the protein molecular weight marker bands (in kDa) is indicated on the right. (**b**) Real-time PCR analysis of CTBP2 expression in MCF-7/Zeb1 cells at 0, 24, and 72 h after doxycycline treatment, respectively. The expression level of CTBP2 in control cells was taken arbitrary as 1.

**Figure 5 molecules-26-03143-f005:**
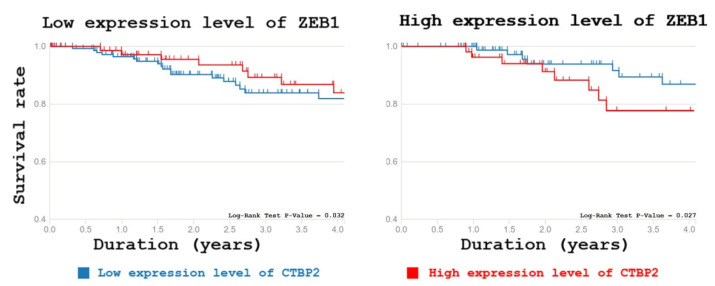
Functional interaction of Zeb1 and CTBP2 affects survival of breast cancer patients. Kaplan–Meier survival rates of breast cancer patients with different levels of CTBP2 and Zeb1 expression demonstrates that high expression levels of both Zeb1 and CTBP2 is associated with poor survival.

**Table 1 molecules-26-03143-t001:** Oligonucleotide sequences used in qPCR.

Gene	5′-3′ Primer Sequence
E-cadherin	F: CTTCTGCTGATCCTGTCTGATG
	R: TGCTGTGAAGGGAGATGTATTG
GAPDH	F: GTCTCCTCTGACTTCAACAGCG
	R: ACCACCCTGTTGCTGTAGCCAA
ZEB1	F: GGCATACACCTACTCAACTACGG
	R: TGGGCGGTGTAGAATCAGAGTC
CtBP2	F: CGTTCTCAGAGCTGGGATGC
	R: TCTGCTGTGCCATACGTCAG
N-cadherin	F: AGCCCGGTTTCATTTGAGGG
	R: TTGAGGGCATTGGGATCGTC
Vimentin	F: TGTCCAAATCGATGTGGATGTTTC
	R: TTGTACCATTCTTCTGCCTCCTG
ERα	F: TGATGAAAGGTGGGATACGA
	R: AAGGTTGGCAGCTCTCATGT

## Data Availability

Data available in a publicly accessible repository. The data presented in this study are openly available in: https://data.mendeley.com/drafts/y6jmyj2fkt (accessed on 21 May 2021).

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
