# Peer review of "Proteomic Analysis of Zeb1 Interactome in Breast Carcinoma Cells"

_molecules, 2021, doi:10.3390/molecules26113143_

Round 1

Reviewer 1 Report

In this manuscript, Mittenberg et al describe proteomic analysis of the EMT-TF Zeb1 expressed exogenously in MCF-7 cells. There are some interesting data in the manuscript, but some validation and experimental details are lacking. I offer the following comments to assist the authors in improving their manuscript.

General comments:

  • Multiple papers have published Zeb1 interactomes using Mass-Spec in various cell lines (eg. Manshouri et al, Nat. Comm. 2019; and Zhang et al, Nat. Cell Biol. 2014). The results from other mass-spectrometric analyses already published should be compared with the results obtained by the authors. Results should be cross-compared to identify common proteins across cell types. These can be included as a table and/or in the discussion.Why is this particular Zeb1 experiment novel? 
  • Justification of the cell line used would help. MCF7 cells do not normally express Zeb1. Would simply expressing Zeb1 induce EMT in the same manner as would occur naturally? Some studies have suggested the other EMT-TFs such as Snail are upregulated prior to Zeb1. Would directly expressing Zeb1 alter the normal course of events in EMT? These points should at least be discussed.
  • The EMT phenotype should be investigated with invasion/ wound healing assay as well as gene expression changes. Some studies have demonstrated that there is a dichotomy between gene expression changes and phenotypic changes, and both are needed for EMT. So a wound-healing or invasion assay would help convince this reviewer that Zeb1 truly induced a complete EMT.
  • Immunoblots in Figure 4 need to be improved significantly. Typically, both IP and unbound (supernatant) fractions, as well as Input is shown. IgG should be included as a control. These blots are not very convincing as IPs.
  • Alternative experiments such as Co-immunofluorescence should be considered to show potential association of the proteins with Zeb1.

Author Response

Dear Reviewer,

Reviewer 2 Report

In the present manuscript authors analyzed Zeb1 interactome in breast carcinoma cells.

I have several suggestions that could be useful to improve the quality of the manuscript: 

  1. Avoid to use abbreviation in abstract section or specify its meaning
  2. ZEB1 is a transcription factor that represses the expression of ERa that play a pivotal role both in breast cancer and in EMT. As a consequence, authors could include the relationship among them in introduction section. They also could include information about triple negative breast cancer or other types of metastatic cancer where EMT is actively involved.
  3. Figure 1 should include a DAPI nuclei staining. Changes in morphology are not really evident. An increase in magnification could be useful. 
  4. In figure 2A there is a cut corresponding to E-cadherin and beta-actin. Samples corresponding to 0,24 and 72h should be run in the same time.
  5. Authors should analyze also other EMT markers as reported by
    https://doi.org/10.1186/s13104-020-05214-y
  6. In figure 2B the statistical analysis is lacking. A western blot analysis of PUMA and BAX is lacking. 
  7. If authors used AACT method to analyze RT_PCR results, why in control cells the expression level is not 1? Y axis  labeling is lacking.
  8. The quality o western blot in figure 4 is vvery and should be increases. Additionally several control are lacking. As an example, total cell lysates results, western blot of Zeb1 should be included.

Author Response

Dear Reviewer,

Round 2

Reviewer 2 Report

reference 36 is lacking.

If possible, increase the magnification of figure 1.

Include representative images of wound healing assay. 

Avoid to use with or without EMT induction, because you only induce the expression of Zeb1 protein.

Blot of vimentin, N-Cadherin and  ERa should be included.

A blot of Zeb1 in CoIp experiments is lacking.

It is already known the ability of Zeb1 to interact with CTBP2.  Disrupting the interaction among the proteins could avoid EMT?

Author Response

We would like to thank the reviewer for insightful comments on our manuscript.

You can find our point-by-point response to the reviewer’s critique in the attached file.
